# Dynamic Induction Heating Temperature Field Analysis of Spiral Bevel Gears

**Yin Zhang [1,2], Hui Zhang [1,2], Yixiong Yan [3,*] and Pengfei Zhu [1,2]**

[1] The State Key Laboratory of High-Performance Complex Manufacturing, Central South University, Changsha 410083, China

[2] College of Mechanical and Electrical Engineering, Central South University, Changsha 410083, China

[3] Department of Mechanical and Electrical Engineering, College of Automotive and Mechanical Engineering, Changsha University of Science and Technology, Changsha 410114, China

[*] Correspondence: yanyixiong1111@163.com

**Abstract:** Combining tooth surface induction heating and shot peening is an efficient method to improve tooth surface performance. Reasonable designs of the induction coil structure and parameters are essential for achieving uniform and efficient tooth surface heating. In this work, to precisely control the tooth surface temperature field and improve the heat uniformity across the tooth surface, a transverse coil (TC) and a longitudinal coil (LC) were designed, and the gear was set to rotate at a constant speed of 20 r/min, dividing the tooth surface is into a shot-peening area and heating area. Further, dynamic numerical simulations were performed using COMSOL Multiphysics in combination with the uniform rotation of gears to investigate the effect of the coil structure on the temperature field of the outer surface of gears. The results of the analyses combining the effects of different electrical parameters revealed that the gear surface temperature under LC heating was more uniformly distributed in the axial and circumferential directions, the tooth surface temperature fluctuations were smaller, the temperature difference between the root and top of the tooth was smaller, an coil heating was more efficient. Thus, the LC was deemed suitable for use as the spiral bevel gear induction heating coil. Finally, heating experiments were conducted using the LC to validate the simulation model. The results show that the use of LC heating can achieve the research goals of uniform temperature field distribution on the tooth surface and efficient temperature rise, providing the prerequisite for shot peening.

**Keywords:** dynamic induction heating; spiral bevel gear; coil structure; temperature field

## 1. Introduction

Spiral bevel gears have the advantages of a high load-bearing capacity, low noise, and smooth transmission, and are widely used in the automotive industry. However, with the increase in vehicle carrying power, fatigue failures of spiral bevel gears are becoming increasingly severe, and high-performance tooth surface strengthening has become an important solution [1]. Shot peening is a common method of tooth face peening that increases tooth face hardness by converting tooth face tensile stress into compressive stress, thereby improving tooth face fatigue and wear resistance [2]. Compared with common shot peening, induction heating shot peening can effectively increase the residual stress on the workpiece surface, refine the grains on it, and improve its dislocation density and hardness [3–5]. Using high-frequency electromagnetic induction heating technology to heat a spiral bevel gear before shot peening can increase the gear temperature and provide conditions for composite shot peening of the tooth surface. Therefore, it is important to study the temperature distribution of the tooth surface during the induction heating of spiral bevel gears.

The most commonly used research method for induction heating problems is based on numerical simulations. While studying the field distribution law of induction heating

temperature, Huang et al. [6] used multiple physical coupling analyses in ANSYS software to predict the temperature distribution of each coil layer, determined the optimal combination of process parameters for two-layer induction coils, and quantitatively analyzed the correlation between coil structure and temperature distribution. To obtain a more uniform temperature distribution, in terms of heating parameter optimization, Wang et al. [7] established a new finite element model of pipeline scanning induction heating, analyzed the main factors affecting the uneven temperature distribution of the lining layer, and used the response surface method to optimize the system parameters.

The structure of the induction heating coil varies with the workpiece shape, and the distance between the induction heating coil and the workpiece affects the heating efficiency and uniformity of the temperature distribution on the workpiece surface. Several studies have considered the effects of coil structure and workpiece geometry during induction heating, and some optimization analyses have been performed. Fu et al. [8] investigated the effect of the coil structure on the temperature distribution of a heated work-piece, and the results showed that a longitudinal magnetic flux coil could obtain a better temperature distribution. Jakubovicova et al. [9] optimized the geometry of the induction coil to obtain a uniform temperature distribution on the workpiece surface.

To study the effect of magnetizer geometry on heating rate and temperature uniformity, Gao et al. [10] investigated the effect of magnet guide geometry on heating rate and temperature uniformity during local induction heating of AISI 1045 steel work-pieces. They verified the results of numerical simulations with experiments to obtain the effect law of the magnet guide on temperature uniformity and the geometry of the magnet guide with the best heating rate. Nian et al. [11] analyzed the effects of the thickness of the heating target, coil turn distance, heating distance, induction coil position, operating frequency, and waiting time on the heating rate and temperature uniformity of induction heating. The results showed that the thickness of the workpiece affects heating rate, and the induction coil position affects heating uniformity.

To investigate the induction heating of gears, Wen et al. [12] studied the moving in-duction heating process of a wind turbine's inner gear ring and reduced the gear ring's temperature inhomogeneity by changing the heating's starting and stopping positions. Zhao et al. [13] analyzed the effect of the number of iterations of heating on temperature distribution uniformity during asynchronous dual-frequency induction heating of bevel gears. The results showed that temperature distribution uniformity in the bevel gear tooth width and tooth profile direction improved with an increase in the number of iterations. Induction heating has been discussed extensively in previous studies; however, induction heating of spiral bevel gears has rarely been discussed.

Here, two structures of induction heating coils were designed for spiral bevel gear geometry and shot peening process requirements. A dynamic induction heating simulation analysis was performed with the spiral bevel gear rotating at a constant speed of 20 r/min. According to the characteristics of tooth surface area division, induction-heating experiments were conducted using longitudinal coils to validate the simulation model [14–16]. This study can be applied to the construction of heating devices in the shot peening of induction heating composite, which provides a reference for selecting the induction coil and a theoretical basis for dynamic induction heating, as well as a prerequisite for induction heating shot peening.

## 2. Modeling of Induction Heating Coils for Spiral Bevel Gears

### 2.1. Induction Coil Modeling

Our induction heating mathematical model establishes an induction coil geometry model for the spiral bevel gear spatial shape characteristics and studies the "electromagnetic-thermal-motion" multi-physical field coupling mechanism of the induction coil and gear [17,18]. In this paper, the pinion is selected as the research object, because the pinion is more prone to tooth surface wear and fatigue failure in the transmission process. Therefore, we consider the temperature field distribution uniformity of the pinion after induction heating as the

research objective to prepare for subsequent shot peening of the tooth surface. The structure of the spiral bevel gear is shown in Figure 1, and relevant parameter information is listed in Table 1.

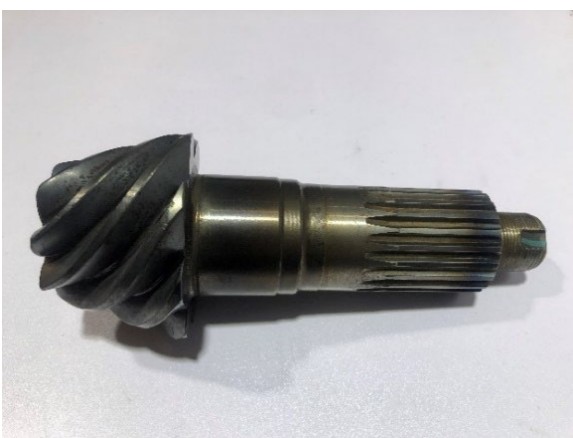

**Figure 1.** Spiral bevel gear structure.

**Table 1.** Spiral bevel gear parameters.

| Parameters | Value |
|---|---|
| Number of teeth | 8 |
| Modulus/m | 6.8610 |
| Helix angle $\beta/°$ | 40° |
| Pressure angle $\gamma/°$ | 22° |
| Tooth width $L_1/$mm | 44 |
| Tooth top height factor | 0.8500 |
| Top gap coefficient | 0.1880 |
| Hand of helix | Right |

The structural design of the heating coil of the spiral bevel gear induction is based on the spiral bevel gear geometry and the constraints of the shot peening equipment. The distance between the induction heating coil and the outer conical surface of the gear, that is, the air gap h, should be considered.

To avoid damage to the induction coil during shot peening, the induction coil must be reserved for an opening in the design process. The coil opening angle is represented by $\theta_0$. Part of the coil near the shot-peening area is bent along the direction of the conical surface generatrix, and its bending height is represented by $H_0$. $d_1$ represents the maximum coverage diameter of the projectile on the tooth surface and the $L_1$ represents the tooth width. The assembly of the spiral bevel gear and induction heating coil is shown in Figure 2.

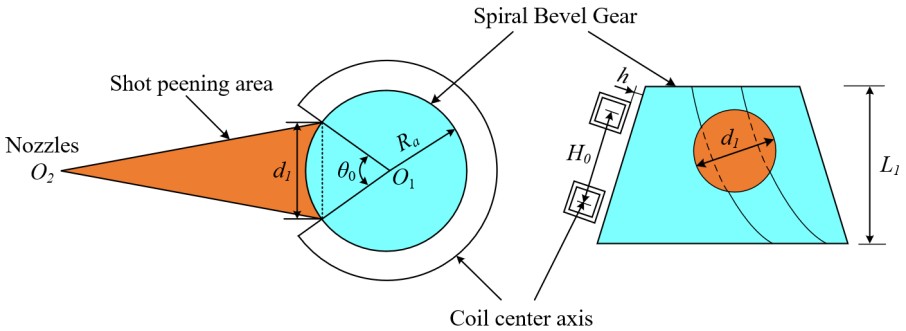

**Figure 2.** Assembly relationship between the spiral bevel gear and induction heating coil.

Based on the relationship between the nozzle and spiral bevel gear assembly in the shot peening process, if the induction coil opening angle is too small, the projectile interferes with the induction coil and damages it. In contrast, if the coil's opening angle is too large, the coverage of the coil on the gear is reduced, and the heating efficiency of the coil decreases. The angle $\theta_0$ of the induction heating coil opening is calculated as follows:

$$\theta_0 \min = \arcsin\left(\frac{d_1/2}{R_a}\right) \tag{1}$$

where $d_1$ is the diameter of the shot-peening coverage area and $R_a$ is the tooth top circle radius of the small end face of the spiral bevel gear.

Combining the calculated basic parameters of the coils, a transverse coil (TC) and longitudinal coil (LC) structures were designed for induction heating shot peening of spiral bevel gears, as shown in Figure 3. The induction heating coil material is metallic copper with a square cross section and a hollow coil interior. A water stream is connected inside the coil to cool the heat generated by the coil conductor during induction heating. The basic parameters of the two coils are shown in Table 2.

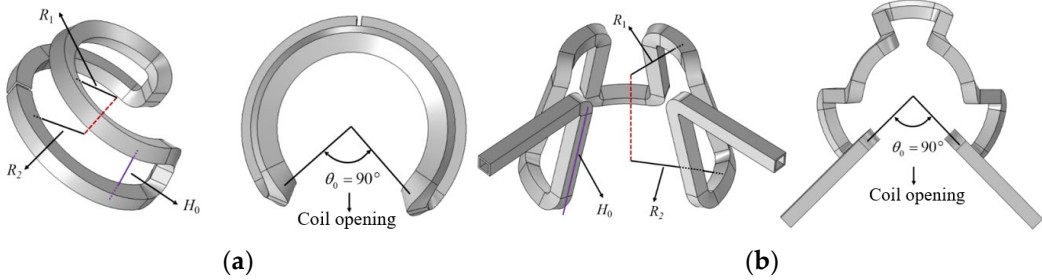

(**a**)                                         (**b**)

**Figure 3.** Two induction coil structures: (**a**) transverse coil, (**b**) longitudinal coil.

**Table 2.** Induction heating coil parameters.

| Parameters | Transverse Coil | Longitudinal Coil |
|---|---|---|
| Materials | Copper | Copper |
| Section size/mm | $8 \times 8$ | $8 \times 8$ |
| Opening angle/° | 90° | 90° |
| Wall thickness/mm | 1.5 | 1.5 |
| Bend height $H_0$/mm | 30 | 60 |

*2.2. Mathematical Model*

The eddy currents induced by the alternating magnetic field generated by the induction coil provide the main heat for the dynamic induction heating of the spiral bevel gear. Therefore, to obtain the temperature model of the spiral bevel gear surface, it is necessary to first calculate the eddy currents induced on the gear surface based on the electromagnetic field generated by the induction coil, and take the heat generated by the eddy work as the internal heat source to obtain the temperature field model of the gear surface. The electromagnetic field characteristics of induction heating can be represented by Maxwell's equations, Ampere's law, which describes the relationship between the current and the direction of magnetic induction lines, Faraday's law of induction which describes the quantitative relationship between the amount of electricity passed on the electric level and the precipitated material, Gauss's law of flux, which describes the magnetic field as a

passive field, and Gauss's law of flux, which describes the relationship between the charge and electric flux. The differential expression is:

$$
\begin{cases}
\nabla_H \times \vec{H} = \vec{J} + \frac{\partial \vec{D}}{\partial t} \\
\nabla_H \times \vec{E} = -\frac{\partial \vec{B}}{\partial t} \\
\nabla_H \cdot \vec{B} = 0 \\
\nabla_H \cdot \vec{D} = \rho
\end{cases}
\tag{2}
$$

where $\nabla_H$ is the hamiltonian, $\vec{H}$ is the magnetic field strength vector, $\vec{J}$ is the current density vector, $\vec{D}$ is the electric induction intensity vector, $t$ is the induction heating tie, $\vec{E}$ is the electric field strength vector, $\vec{B}$ is the magnetic induction strength and $\rho$ is the charge density.

When applying Maxwell's system of equations, the electromagnetic field quantities and the gear steel must satisfy the following intrinsic relationship equation:

$$
\begin{cases}
\vec{D} = \varepsilon \vec{E} \\
\vec{B} = \mu \vec{H} \\
\vec{J} = \sigma \vec{E}
\end{cases}
\tag{3}
$$

where $\varepsilon$ is the dielectric constant of gear steel, $\mu$ is the relative magnetic permeability of gear steel, and $\sigma$ is the electrical conductivity of gear steel.

In the dynamic induction heating process of a spiral bevel gear, the heat generated by the induction eddy current as the internal heat source heats the surface layer of the gears rapidly, while a part of the heat is transferred to the core of the spiral bevel gears and the gear shaft by heat conduction. The remaining part of the heat is dissipated into the surrounding environment by heat radiation and heat convection. Gear steel is an isotropic material, and the heat transfer control equation of its three-dimensional transient temperature field is:

$$
c(T)\rho(T)\frac{\partial T}{\partial t} = \frac{\partial}{\partial x}\left(\lambda(T)\frac{\partial T}{\partial x}\right) + \frac{\partial}{\partial y}\left(\lambda(T)\frac{\partial T}{\partial y}\right) + \frac{\partial}{\partial z}\left(\lambda(T)\frac{\partial T}{\partial z}\right) + Q
\tag{4}
$$

By introducing the Laplace operator $\nabla_L$, the above equation can be simplified to:

$$
c\rho_m \frac{\partial T}{\partial t} = \lambda \nabla_L^2 T + Q
\tag{5}
$$

where $c$ is the specific heat capacity of gear steel materials, $\rho_m$ is the density of gear steel materials, $\lambda$ is the thermal conductivity, $T$ is the gear temperature and $Q$ is the joule thermal power density.

The gear convection and radiation boundary condition equations are:

$$
\lambda \frac{\partial T}{\partial n} = -k(T - T_m) - \varepsilon_a \sigma_a (T - T_m)
\tag{6}
$$

where $k$ is the heat transfer coefficient between gear surface and air, $T_{in}$ is the ambient temperature, $n$ is the external normal to gear surface, $\varepsilon_a$ is the gear surface emissivity and $\sigma_a$ is the Stefan-Bozeman constant.

### 2.3. Induction Heating Modeling

A three-dimensional model of the spiral bevel gear induction heating with two coil structures was established using COMSOL finite element software (Figure 4). The coil structure consisted of a spiral bevel gear, an air domain, and an induction coil. The yellow arrow in the figure indicates the induction coil's current flow direction, while the white

arrow indicates the gear's rotation direction. The gear area between the red dashed lines is the coil opening area, used as the projectile coverage area during the gear shot-peening process.

Transient analysis was solved using the magnetic field module and the solid heat transfer module of COMSOL finite element software. Tetrahedral mesh cells were used for the induction coil and air domain parts. To reduce the calculation time, the mesh near the induction heating coil was finely divided, and the mesh far from the coil was coarsely divided. The spiral bevel gear was meshed with a hexahedral mesh cell, the tooth surface was meshed with a rectangular mapping mesh, and the mesh inside the gear was in-creased gradually along the shaft diameter direction of the gear center. To improve the accuracy of the temperature solution, the boundary layer mesh of the tooth surface was di-vided into three layers [18,19].

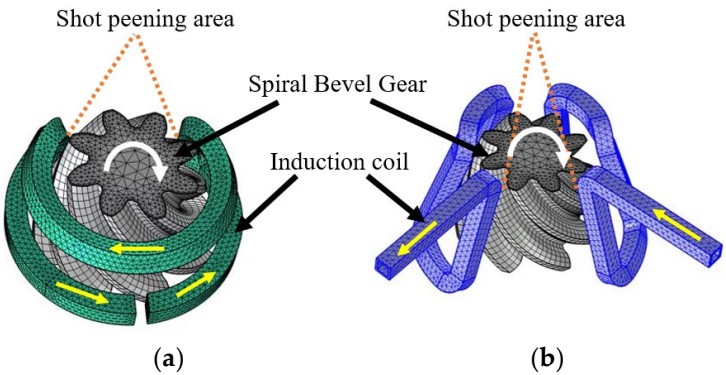

**Figure 4.** Spiral bevel gear induction heating model: (**a**) transverse coil, (**b**) longitudinal coil.

The spiral bevel gear was made of 20CrMnTi gear steel. 20CrMnTi is a material commonly used to manufacture transmission gears in the automotive field, and has high hardenability, high strength and impact toughness. Its chemical composition is shown in Table 3. The electrical and thermal parameters related to the temperature are shown in Figure 5. The spiral bevel gear was set to rotate at a constant speed of 25 r/min around the center of rotation; the induction heating parameters were a current of 1000 A, current frequency of 15 kHz, heating time of 120 s and an air gap of 3 mm.

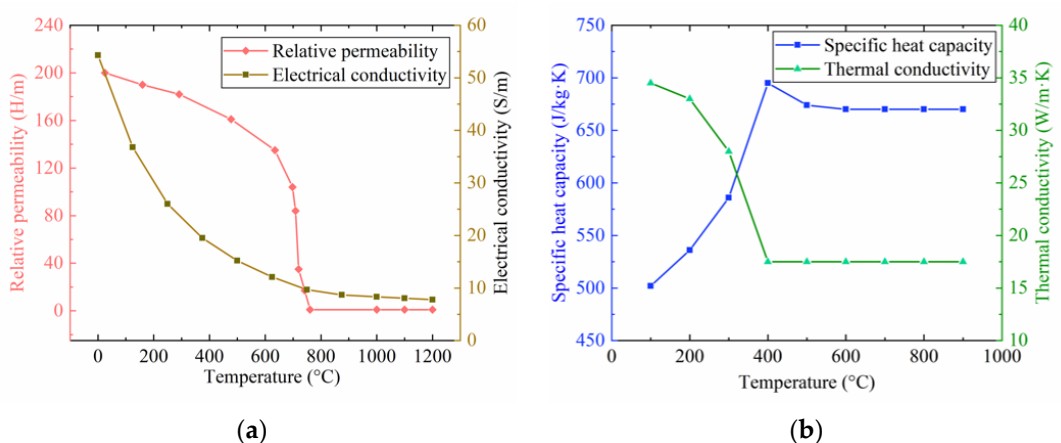

**Figure 5.** Parameter curves of the 20CrMnTi with temperature: (**a**) thermal parameters, (**b**) electrical parameters.

**Table 3.** Chemical composition of 20CrMnTi (wt%).

| C | Si | Mn | P | S | Cr | Mo |
|---|---|---|---|---|---|---|
| 0.19–0.25 | 0.17–0.37 | 0.55–0.9 | ≤0.03 | ≤0.03 | 0.85–1.25 | 0.35–0.45 |

In the electromagnetic field solution process, magnetic insulating boundary conditions are set on the outer surface of the air domain to ensure that the magnetic lines of force are parallel to the outer surface of the air domain. The boundary conditions of heat convection and heat radiation were considered in the simulation analysis of the temperature field of the spiral bevel gear. The convective heat transfer coefficient was set to 10 W/(m·K), radiation rate was 0.7, and external ambient temperature was set to 293 K.

## 3. Simulation Results and Discussion

Induction heating shot peening can effectively improve the stability of residual stresses on the tooth surface, increase the depth of the residual stress strengthening layer, and improve the fatigue performance of the gear. However, the gear steel material undergoes local creep because of the high temperature, causing stress relaxation, which weakens the effect of hot shot peening; thus, the gear at the coil opening shot peening regional temperature is an important control parameter. The unpeened area was below the induction coil and was the main area for induction heating. Therefore, the temperature distribution in the heated area was controlled to prevent overheating. A temperature difference between the gear heating area and the shot peening area was introduced to measure the temperature distribution of the entire spiral bevel gear surface. If the gear surface temperature difference is small, the temperature distribution of the entire spiral bevel gear is more uniform. When analyzing the influence of current on the temperature field, the frequency was set to 15 kHz as a constant value, and the values of the current were 500, 750, 1000, 1250, and 1500 A, respectively. In the analysis of the influence of frequency on the temperature field, the current was set as a constant value of 1000 A, and the frequency values were 10, 12.5, 15, 17.5, and 20 kHz respectively. The main parameters of the simulation are shown in Table 4.

**Table 4.** Simulation parameters.

| Parameters | Transverse Coil | Longitudinal Coil |
|---|---|---|
| Thermal radiation coefficient | 0.7 | 0.7 |
| Initial temperature/°C | 20 | 20 |
| Frequency/kHz | 10, 12.5, 15, 17.5, 20 | 10, 12.5, 15, 17.5, 20 |
| Current/A | 500, 750, 1000, 1250, 1500 | 500, 750, 1000, 1250, 1500 |
| Heating time/s | 90 | 90 |

*3.1. Effect of Current Intensity on Gear Shot-Peening Area and Surface Temperature Difference*

The temperature of the shot peening area and the coil heating area was extracted, the coil covered area was the heating area, and the opening area was the shot peening area, as shown in Figure 6. The research object was to determine the average temperature of the shot peening area (indicated by GSPA) and the average temperature difference between the shot peening area and the heating area (indicated by GSTD).

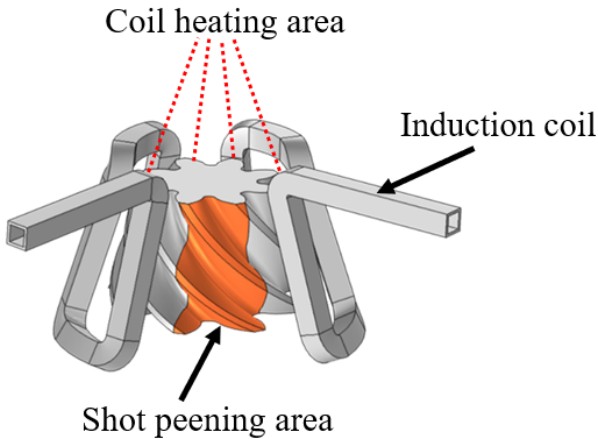

**Figure 6.** Division of gear research area.

We calculated the temperature difference between the gear shot-peening area and the spiral bevel gear surface at different electric currents. The changes in the gear shot peening area (GSPA) temperature and gear surface temperature difference (GSTD) under different input electric currents of the two coil structures are shown in Figure 7. The effect of the electric current on the GSPA temperature is similar in pattern, and the two curves are approximately parallel. At the same electric current, the heating temperature of the TC was approximately 45 °C higher than that of the LC. As for the GSTD, when heated with a TC with a current of 1500 A, the maximum GSTD was 145 °C. The GSTD was approximately 75 °C when heated with an LC. This result indicates that when the gear is heated with a TC, the heat generated by the induced eddy currents is mainly concentrated on the gear surface, resulting in a greater temperature difference and a more pronounced temperature field on the tooth surface. Although the LC has a lower heating power because the coil is approximately parallel to the helical direction of the gear teeth, the tooth surface is heated more uniformly, resulting in a smaller temperature difference and more uniform temperature field on the tooth surface.

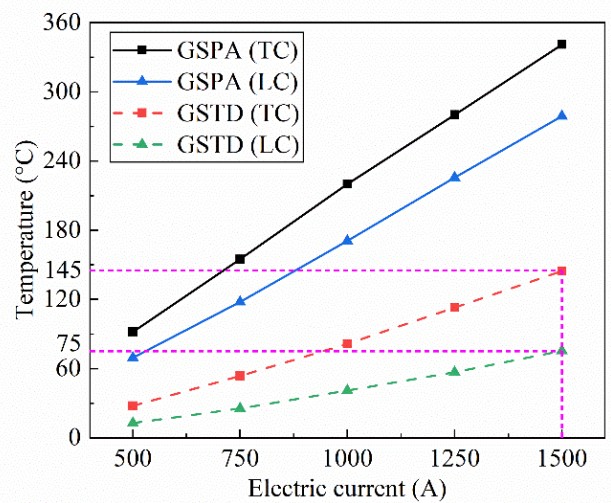

**Figure 7.** Effect of current intensity on temperature distribution.

*3.2. Effect of Current Frequency on Gear Shot-Peening Area and Surface Temperature Difference*

The coil current frequency for the TC was positively correlated with the GSPA and GSTD (Figure 8). In the induction heating process, the depth of current penetration is related to the current frequency; the higher the current frequency, the smaller the depth of current penetration, which is mainly concentrated on the gear surface, significantly

in-creasing the GSPA and GSTD. Conversely, for the LC, the higher the current frequency, the smaller the increase in GSPA and GSTD during induction heating.

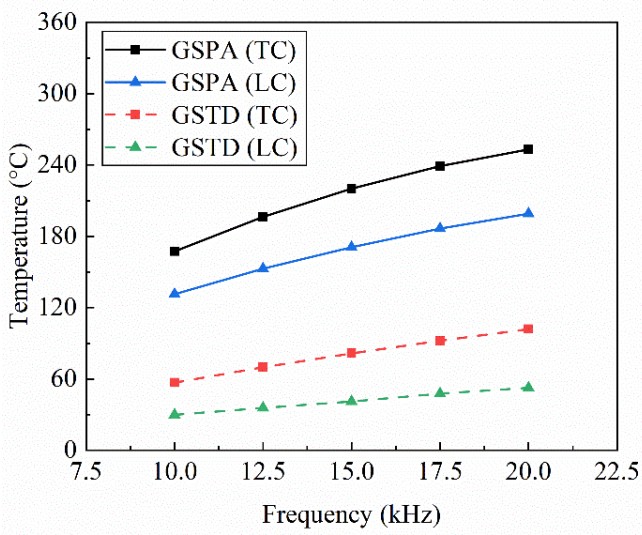

**Figure 8.** Effect of current frequency on temperature distribution.

Comparing TC and LC, the effect of current frequency on the temperature of the spiral bevel gear shot peening area was similar in pattern, and the two curves were approximately parallel. At the same current frequency, the temperature in the GSPA was approximately 50 °C higher with a TC rather than with an LC. When the TC was used for heating with a current frequency of 20 kHz, the temperature difference on the gear surface reached a maximum of 102 °C, whereas when the LC was used for heating, the temperature difference on the gear surface was approximately 50 °C. The temperature values are shown in Table 5. Therefore, with the same electrical parameters, the temperature difference generated on the gear surface was higher when the TC was used for heating. In contrast, the temperature field was more uniform when the LC is used for heating.

**Table 5.** Temperature values.

| Frequency/kHz | GSPA (TC)/°C | GSPA (LC)/°C | GSTD (TC)/°C | GSTD (LC)/°C |
|---|---|---|---|---|
| 10 | 167.3 | 131.2 | 56.8 | 29.7 |
| 12.5 | 196.3 | 152.7 | 69.8 | 35.6 |
| 15 | 220 | 170.7 | 81.4 | 41 |
| 17.5 | 239 | 186.4 | 92.1 | 47.6 |
| 20 | 253 | 198.9 | 101.8 | 52.5 |

### 3.3. Effect of Coil Structure on the Heating Rate of the Gear Shot-Peening Area

One point each on the tooth top (TT) and tooth root (TR) of the spiral bevel gear in the middle of the shot-peening area was selected as the study object. This selection accurately analyzed the tooth surface temperature of the spiral bevel gear during the induction heating process, thereby ensuring the uniformity of the tooth surface temperature in the gear shot-peening area (coil opening area). The electrical parameters were set to 15 kHz and 1000 A, and the temperature distribution data from the simulation results were extracted (Figure 9).

Spiral bevel gear shot peening area with TC heating to 170 °C took 85 s, whereas the LC heating took 118 s. During induction heating, with the rotation of the spiral bevel gear, the research target point periodically passed through and away from the shot peening area of the coil opening. Hence, the temperature curve of the study target point on both the top and root of the tooth changed in a periodic fashion, and the temperature repeatedly increased and decreased. When the research target point was in the coil opening area, the

magnetic field strength near the research point decreased. Thus, the electromagnetic heat generated in this area was lower than the energy dissipated at this time, resulting in a lower temperature. When the research target point was far from the coil opening area and located below the heating coil, the magnetic field strength near the research point significantly enhanced again. Thus, the electromagnetic heat at the point was significantly increased again, resulting in a temperature increase.

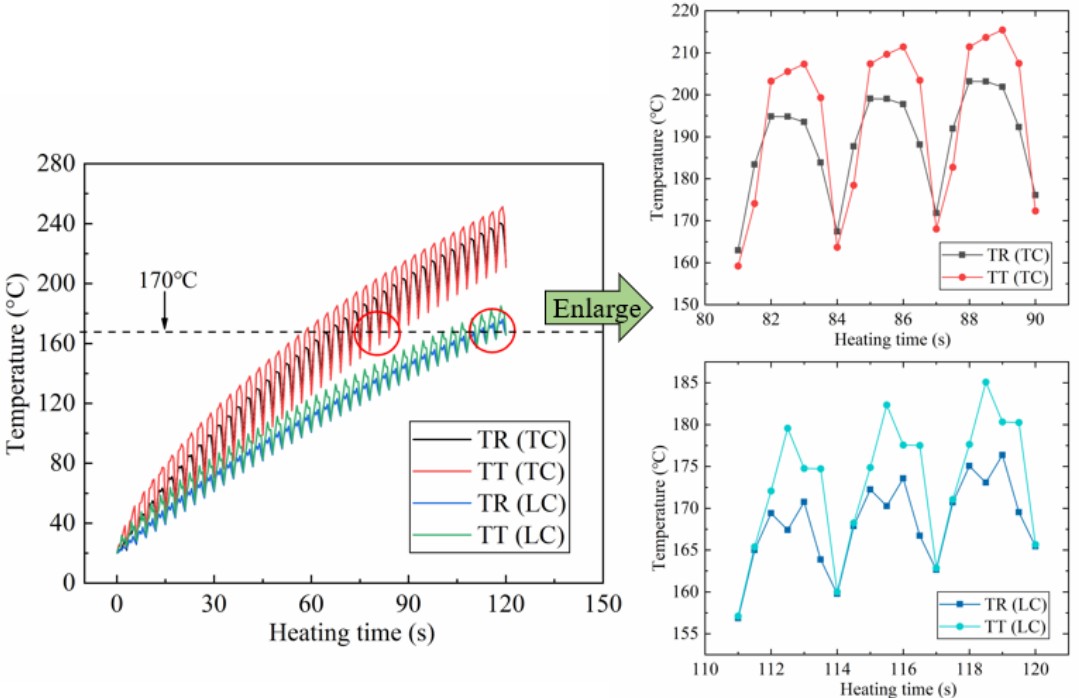

**Figure 9.** Temperature rise process under two coil structures.

The magnitude of the temperature fluctuations at the study point also reflects the heating efficiencies of the two coils. The small temperature fluctuations indicate that the temperature of the tooth surface point does not drop significantly within one cycle of gear rotation, and the heating efficiency is high. According to Figure 9, in one rotation cycle, with the same electromagnetic parameters, the temperature fluctuation of the gear heated by the horizontal coil is approximately 35 °C, and the tooth root temperature in the shot peening area is 5 °C lower than the tooth top. The temperature fluctuation of the vertical coil heated gears was 15 °C, and the root and top temperatures in the shot peening area were approximately the same.

### 3.4. Effect of Coil Structure on Gear Tooth Surface Temperature Distribution

Because of the complex shape of the spiral bevel gear tooth surface, to study the temperature distribution law of the spiral bevel gear tooth surface, five circumferential curves were selected in the axial position of the tooth surface, namely, A, B, C, D, and E. Each curve represents the outer surface profile in the gear tooth width direction. The plane where each curve was located was parallel to each other with a spacing of 11 mm, and 16 points on each curve were taken as the study object (Figure 10).

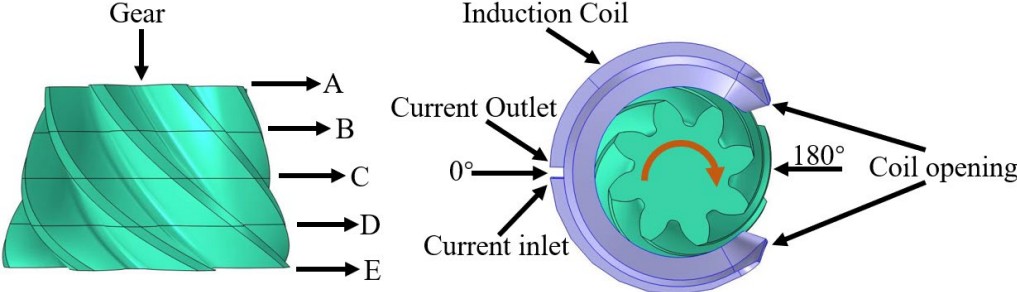

**Figure 10.** Selection of gear surface study curves.

The axial temperature distribution on the spiral bevel gear surface after induction heating using a TC is shown in Figure 10. When the gears were heated using TC, the circumferential temperature distribution of the tooth surfaces was not uniform. The five temperature curves in the figure show periodic changes as the temperature increases and decreases. For example, between 250° and 300°, the temperature curves A, C, and E increase and then decrease, whereas curves B and E decrease and then increase. Between 250° and 300°, the temperature curves A, C, and E increase and then decrease, whereas curves B and E decrease and then increase. This is due to the difference in the shapes of the spiral bevel gears with different curves at the same angle. The teeth of the spiral bevel gears have a helix angle, so there are different tooth tops and roots at the same angle. There is a radial distance difference between the tooth root and top. When the tooth top was close to the heating coil, the electromagnetic heating power in this area was high, and the temperature was high. When the tooth root is far from the heating coil, the electromagnetic heat power in this area is low, and the corresponding temperature is low.

According to the results of TC heating (Figure 11), curves A, B, C, D, and E have the lowest temperature at the coil opening from 135° to 225°. In contrast, the temperature farther away from the coil opening was higher, and the distribution was more uniform. The temperature distribution in the gear axial direction was uneven. The B, D, and E curves correspond to higher temperatures because the gear area where these three curves are located is closer to the induction heating coil. Conversely, the gear area where the A and C curves are located is farther away from the induction heating coil, and their corresponding temperatures are lower.

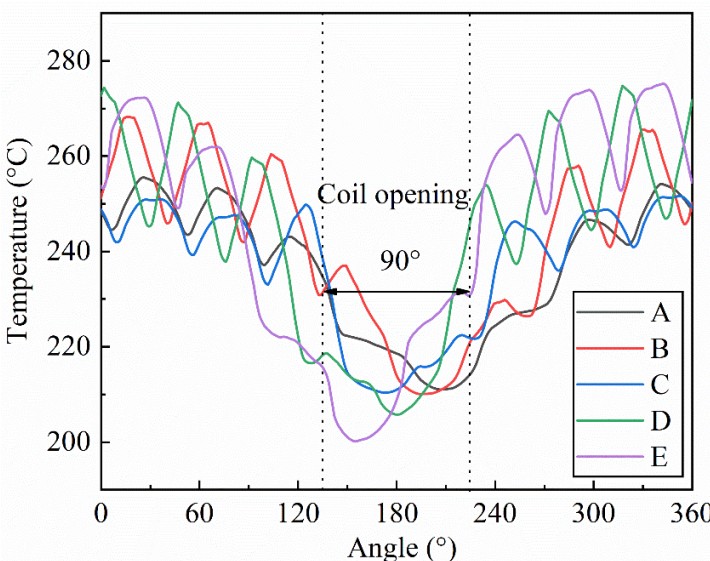

**Figure 11.** Temperature distribution of the gear tooth surface during TC heating.

As illustrated in Figure 12, when using an LC to heat the spiral bevel gear, the temperature distribution around the gear's circumference was similar to the results of the TC heating, and the temperature curve varies periodically. The temperature was the lowest at the coil opening from 135° to 225°, the highest tooth surface temperature was located near 90° and 270°, the coil was roughly parallel to the tooth direction of the gear, and the efficiency of induction heating was the highest. In the axial direction, the LC was more uniformly heated than the TC, and the temperature difference between the five curves was smaller.

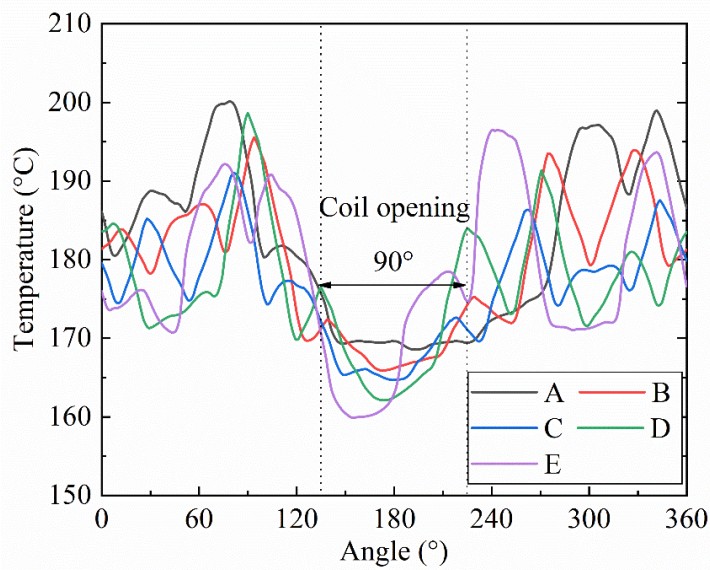

**Figure 12.** Temperature distribution of the gear tooth surface during LC heating.

### 3.5. Electromagnetic Heat Distribution Law of Tooth Surface

The temperature distribution trend of the tooth surface can be obtained by the distribution law of the electromagnetic heat curve. Therefore, to analyze the effect of the coil structure on the temperature distribution of the spiral bevel gears in the circumferential and axial directions, the distribution law of electromagnetic heat on the gear surface when heated by the two types of coils was investigated. The electromagnetic heat distributions of the gears heated by the two coils are shown in Figure 13.

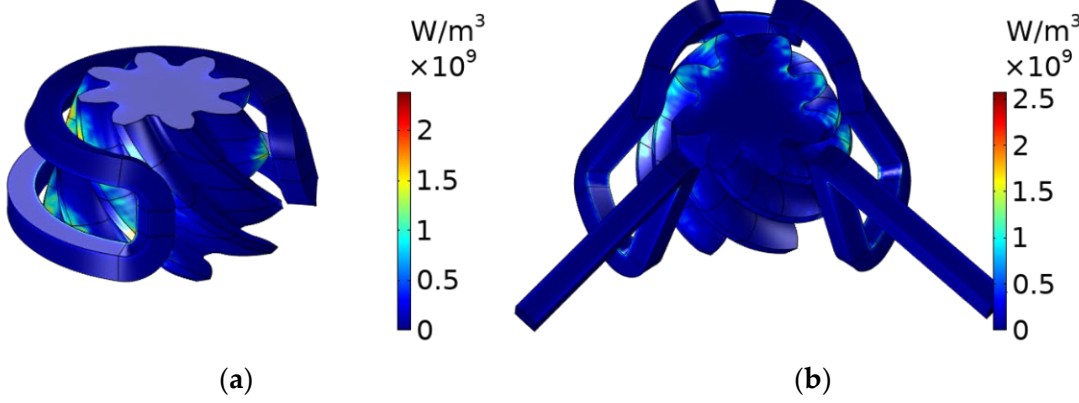

**Figure 13.** Cloud diagram of electromagnetic heat distribution during heating of two coil structures: (**a**) TC heating, (**b**) LC heating.

Due to the skin effect of induction heating, the electromagnetic heat generated by TC heating was mainly on the outer surface of the spiral bevel gear. Because of the radial distance between the root and top of the gear, the electromagnetic heat was unevenly distributed in the circumferential and axial directions in the area of the tooth surface below the coil. Therefore, the electromagnetic heat was highest in the tooth top area near the coil and gradually decreased towards the tooth root area.

As shown in Figure 14, because the small end face of the gear was far away from the induction coil, the surrounding electromagnetic heat was distributed sparsely, such that the electromagnetic heat on curve A was significantly lower than that of the other curves. Hence, the electromagnetic heat concentration distribution areas in the circumferential direction of the tooth surface were located at 0–135° and 225–360°, whereas in the gear shot-peening area at the coil opening, the electromagnetic heat suddenly decreased and tended to zero. This explains the temperature drop in the shot-peening area of the gear and the variation law of the circumferential temperature of the tooth surface when the TC was heated. Comparing the five curves of electromagnetic heat distribution, we can see that the electromagnetic heat is sparsely distributed on curves A and C and densely distributed on curves B, D, and E. This distribution is similar to the circumferential temperature distribution of the tooth surface when the TC was heated.

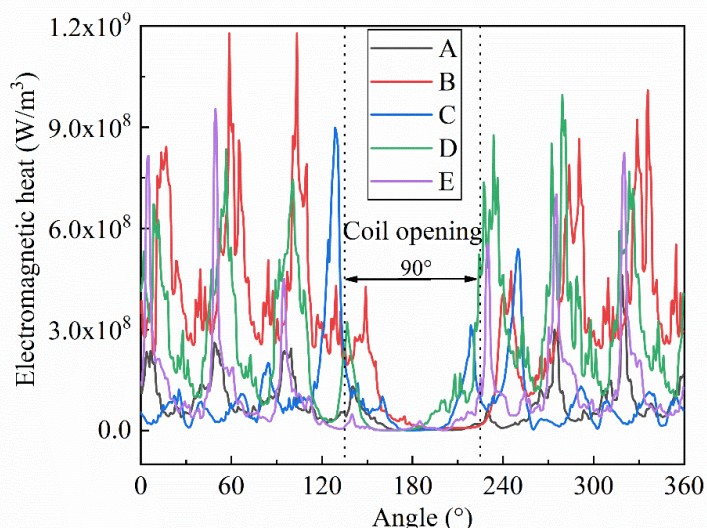

**Figure 14.** Electromagnetic heat distribution curve of the tooth surface under TC heating.

The electromagnetic heat distribution law generated using LC heating differed from that generated through TC heating (Figure 15). In the axial direction of the gear, the electromagnetic heat distribution on the five curves was more uniform. This is because the horizontal part of the coil was distributed on the gear's small end face, and the electromagnetic heat on the gear's small end face was close to that of the other parts of the tooth surface. The electromagnetic heat shown on curve A is similar to that of several other curves, and the temperature difference on the tooth surface was smaller than that when heated by the TC. In the circumferential direction of the gear, the electromagnetic heat at the coil opening abruptly changed to 0, the temperature of the tooth surface started to decrease, and the temperature of the tooth surface increased again when entering the coil coverage area.

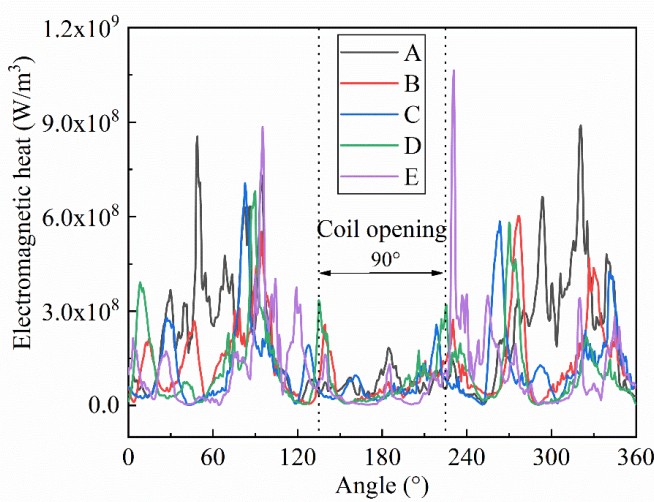

**Figure 15.** Electromagnetic heat distribution curve of the tooth surface under LC heating.

## 4. LC Heating Experimental Test

According to the simulation analysis results in Section 3, the temperature difference of the gear tooth surface was smaller when LC was used for heating, and the temperature field distribution in the shot peening area was more uniform. Therefore, an induction heating test bench was built to verify the reliability of heating using LC and the accuracy of the tooth surface temperature distribution [20–22]. Dynamic induction heating tests were performed on a spiral bevel gear with a modulus of 6.810. Infrared thermal imaging equipment was used to detect and record the tooth surface temperature data during the LC heating process. The recorded experimental data were then imported into Smart View software for processing, and the temperature data were compared with the numerical simulation.

### 4.1. Dynamic Induction Heating Experimental Equipment

To improve the uniformity of tooth surface temperature field distribution under coil heating, spiral bevel gears need to rotate at a uniform speed during induction heating. Therefore, the longitudinal coil was fixed in the experiment, and the spiral bevel gear was set at a constant speed of 20 r/min. The induction heating experimental equipment included an insulated gate bipolar transistor (IGBT)-type power cabinet host, output transformer, induction coil, infrared thermal imager, water pipes (cooling effect), and a spiral bevel gear (uniform rotation; Figure 16).

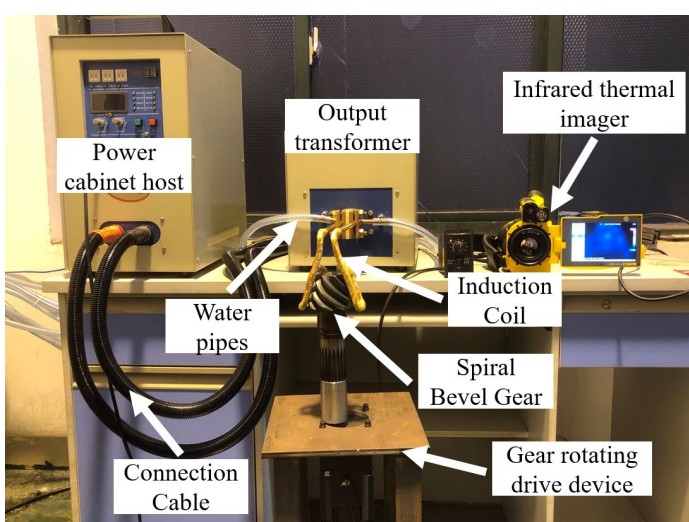

**Figure 16.** Electromagnetic induction heating system.

### 4.2. Selection of Study Points for Spiral Bevel Gears

The top midpoint A and root midpoint B of a gear tooth in the middle section of the spiral bevel gear were selected as the research objects, as shown in Figure 17. This selection helped to accurately determine the tooth surface temperature of the spiral bevel gear during induction heating. In Figure 17, point O is the center point of the gear in this section, path L1 represents the distance from the top midpoint to the center of the gear, and path L2 represents the distance from the root midpoint to the center of the gear.

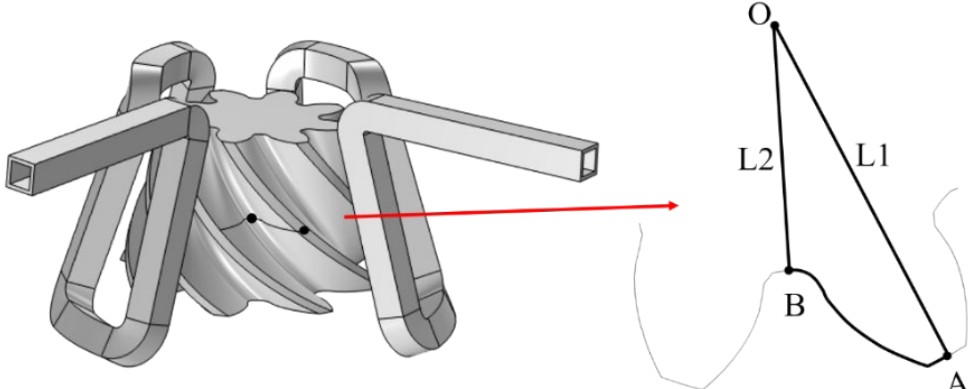

**Figure 17.** Schematic diagram of tooth root and tooth top research points.

### 4.3. Analysis and Comparison of Eexperimental Results

According to the experimental operation procedure, a dynamic induction heating test of the spiral bevel gear was carried out. The experimental data were analyzed in combination with the simulation results. Cloud plots of the experimental temperature distribution (ETD) and numerical simulation temperature distribution (NSTD) at different heating times are shown in Figure 18.

Figure 18 shows that the temperature distribution of the tooth surface in the spiral bevel gear shot peening area exhibits the same trend during the experiment and simulation. It shows a higher temperature at the top of the tooth in the shot peening area, a lower temperature at the tooth root, and the highest temperature at the tooth surface profile. This is because the radial distance between the tooth root and the induction coil is greater than that of the tooth top during induction heating. Owing to the proximity effect, there is less magnetic leakage loss and more efficient heating at the top of the tooth than at the root, so the temperature is higher. Simultaneously, when the gear rotates through the shot peening area, the magnetic induction intensity decreases, and the power of heat dissipation of the gear teeth is greater than the power of heating, which decreases the temperature of the gear teeth. Thus, the top of the tooth is more easily affected by the surrounding airflow, and the temperature drops faster, lowering the top temperature than the tooth surface temperature.

During the heating experiment, the temperature of the induction coil did not change significantly. This is because there was a circulating water flow in the coil during the experiment, and the joule heat generated on the coil was carried away by the water flow, which further illustrates the feasibility of simplifying the water flow during the numerical simulation.

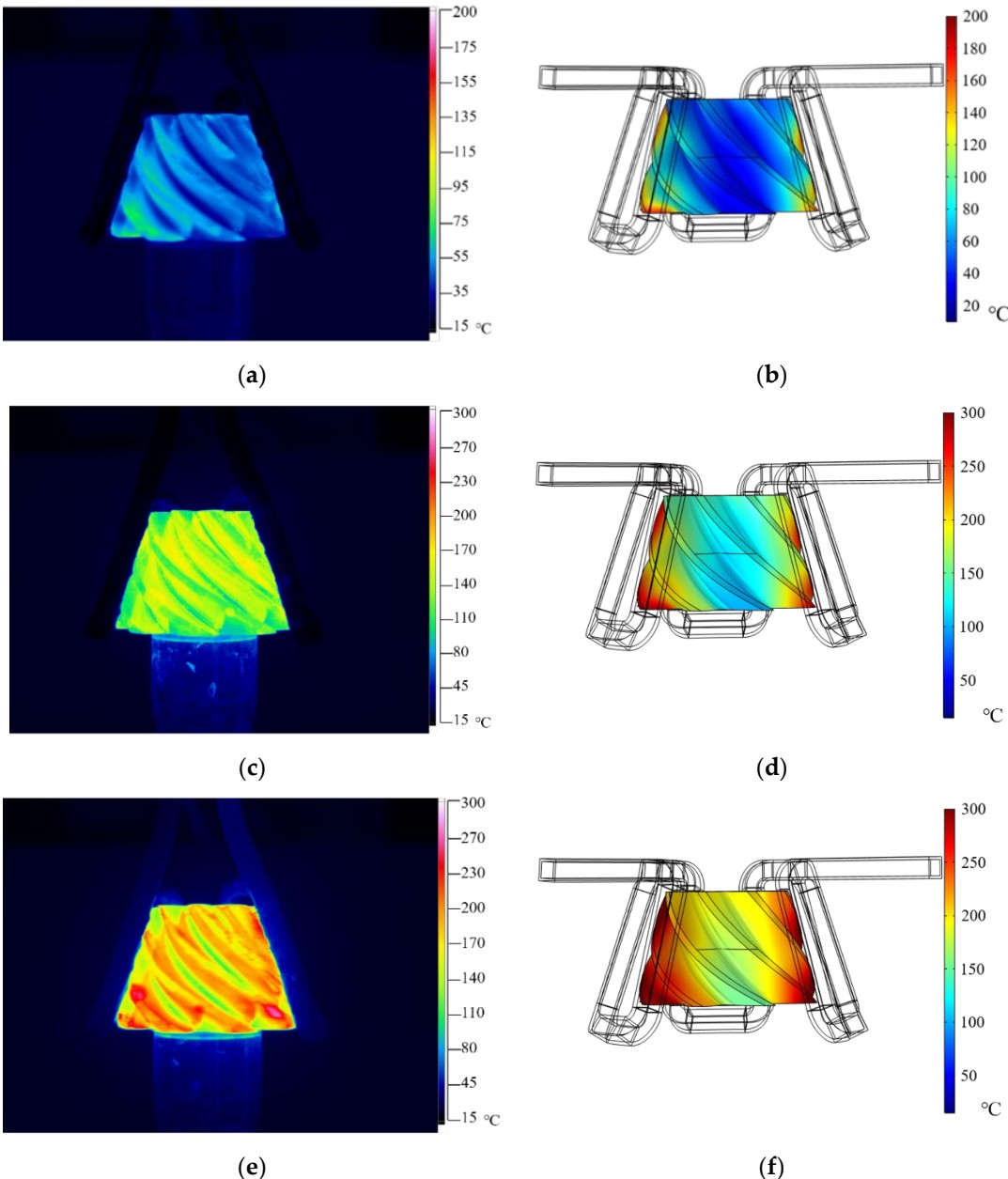

**Figure 18.** Gear tooth surface temperature distribution: (**a**) heating for 3 s in the experiment; (**b**) heating for 3 s in the simulation; (**c**) heating for 18 s in the experiment; (**d**) heating for 18 s in the simulation; (**e**) heating for 30 s in the experiment, and (**f**) heating for 30 s in the simulation.

### 4.4. Comparison of the Experimental and Simulation Results

During the experiment, because the ambient temperature changed with time, the fluctuation of the ambient temperature affected the temperature of the tooth surface, and the infrared thermal imager had a measurement error of 1.5%. To improve the accuracy of the experimental results we conducted the same experiment three times, and the final experimental results are presented as the average of the three experiments. The main purpose of this section is to verify the correctness of the simulation through experiments. Therefore, to quantitatively analyze the error between the experiment and the simulation, the temperature changes of tooth apex point A and tooth root point B during the numerical simulation and experiment of induction heating were collected. The error between the experiment and simulation was compared, and temperature curves were plotted (Figure 19).

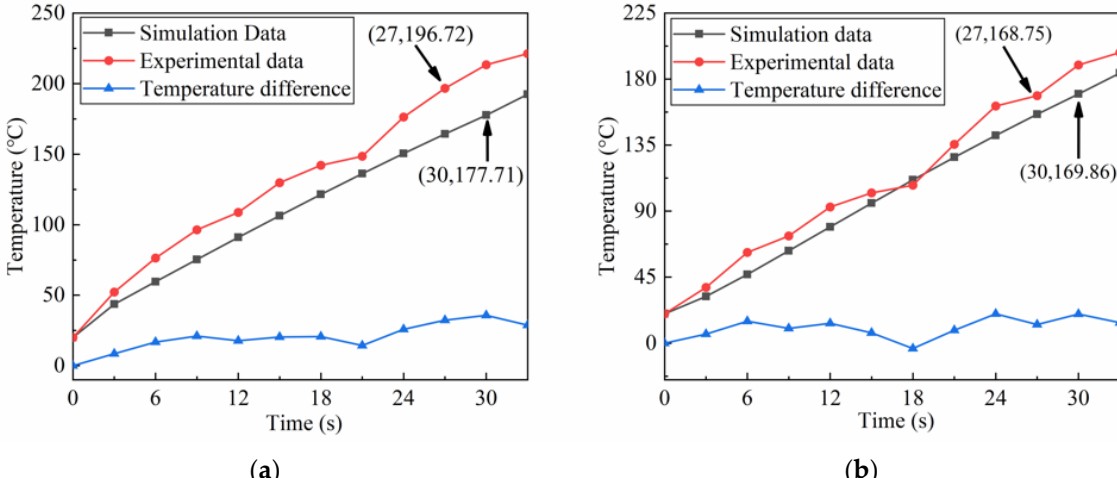

**Figure 19.** Comparison of the experimental and simulated temperatures: (**a**) tooth top point A; (**b**) tooth root point B.

From the temperature curve in Figure 19, the temperatures at points A and B on the tooth surface of the spiral bevel gear show an increasing trend during the heating process, and the curve trends of the simulated and experimental temperatures are the same. How-ever, the experimental temperature is higher than the simulated temperature, and the experimental temperature is more volatile than the simulated temperature. This is because the gear exchanged heat with the air during the experiment, and the temperature of the tooth surface was radiated to the induction coil, resulting in changes in the physical properties of the coil.

As shown in Figure 19a, the average temperature changes during the induction heating of the spiral bevel gear were 5.23 °C/s and 6.1 °C/s. The maximum temperature difference between the simulated and experimental groups at the top of tooth A was 35.75 °C, and the average temperature difference was 20.20 °C, with a maximum relative error of 16.75%. It took 27 s to heat the tooth root point B to 170 °C in the experiment and 30 s to reach the corresponding temperature in the simulation, and the relative error of heating time was 10%. The average temperature change between the simulation and experiment for point B was 4.97 °C/s and 5.4 °C/s. The maximum temperature difference between the simulation and experiment was 20 °C, the average temperature difference was 10.30 °C, and the maximum relative error was 12.36%.

The temperature difference between top A and root B of the spiral bevel gear is shown in Figure 20. The temperature difference between the root and the top of the tooth in the simulation was relatively stable, with an average temperature difference of approximately 9.5 °C, while the temperature difference between the top and the root of the tooth in the experiment fluctuated more, with a maximum temperature difference of 27.3 °C and an average temperature difference of 19.24 °C. This is because of the assumptions of the simulation conditions, which did not consider the influence of the surrounding air and the coil temperature change, the experimental process's external environment, and the measuring instrument's conditions. Overall, the experimental and simulated data for the heating phase are in good agreement.

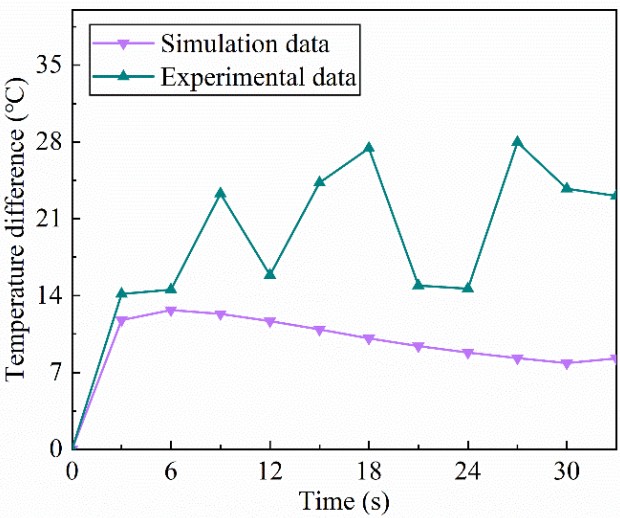

**Figure 20.** Temperature difference between points A and B.

## 5. Conclusions

According to the characteristics of tooth surface area division, dynamic induction heating of gears was carried out, and the research results show that the research objectives of a uniform temperature field and high heating efficiency could be achieved. By comparing the simulation analysis results of the transverse and longitudinal coils through the dynamic induction heating of spiral bevel gears, the longitudinal coil had a more uniform temperature field distribution on the tooth surface after heating than the transverse coil, and the longitudinal coil was determined as the heating coil. Furthermore, the accuracy of the simulation results was verified by designing a longitudinal coil heating experiment. The conclusions are as follows.

(1) The heating efficiency of transverse and longitudinal coils and temperature field distribution of the spiral bevel gear tooth surface at different current frequencies and intensities were investigated. The results show that the surface temperature of the spiral bevel gears heated by the longitudinal coil was more uniformly distributed in the axial and circumferential directions. The temperature fluctuations and temperature differences on the gear surface were smaller, and the temperature differences between the root and the top of the tooth in the shot peening area were also smaller. Therefore, a longitudinal coil was used as the heating coil for the spiral bevel gear.

(2) A spiral bevel gear induction heating test bench was built to conduct the dynamic induction heating experiments. The temperature variation of the spiral bevel gear tooth surface was recorded using a thermal imaging instrument. The experimental results were compared and analyzed using the simulated temperature data. The results show that the average temperature changes of tooth root and tooth top in the experimental measurement were 5.4 °C/s and 6.1 °C/s, respectively. The corresponding temperature changes in the numerical simulation were 5.23 °C/s and 4.97 °C/s. The tooth surface temperature in the experiment and the temperature fluctuation were larger than those in the numerical simulation. This is because the airflow rate and air temperature influenced the tooth surface temperature in the experiment. Overall, the tooth surface temperature distribution in the experiment was consistent with the simulation results. Therefore, according to the research results, the use of longitudinal coils for dynamic heating in the shot peening process can ensure uniform distribution of the tooth surface temperature field, and promote the strengthening effect of the shot peening on the gear.

**Author Contributions:** Y.Z.: Conceptualization; formal analysis; methodology; investigation; writing. H.Z.: data curation; investigation; resources; formal analysis; writing. Y.Y.: funding acquisition; investigation; methodology; validation. P.Z.: data curation; investigation. All authors have read and agreed to the published version of the manuscript.

**Funding:** This research is supported by National Natural Science Foundation of China (Grant No. 52075552).

**Institutional Review Board Statement:** Not applicable.

**Informed Consent Statement:** Not applicable.

**Data Availability Statement:** The author declared that our data and material are available.

**Conflicts of Interest:** The authors declare that they have no conflict of interest in this work. We declare that we do not have any commercial or associative interest that represents a conflict of interest in connection with the work submitted.

## Nomenclature

| | |
|---|---|
| $\theta_0$ | Angle of the coil opening |
| $d_1$ | Diameter of blast coverage area |
| $R_a$ | Radius of the tooth top circle of the small end face of the gear |
| $\nabla_H$ | Hamiltonian |
| $\overrightarrow{H}$ | Magnetic field strength vector |
| $\overrightarrow{J}$ | Current density vector |
| $\overrightarrow{D}$ | Electric induction intensity vector |
| $t$ | Induction heating time |
| $\overrightarrow{E}$ | Electric field strength vector |
| $\overrightarrow{B}$ | Magnetic induction strength |
| $\rho$ | Charge density |
| $\varepsilon$ | Dielectric constant |
| $\mu$ | Relative magnetic permeability |
| $\sigma$ | Electrical conductivity |
| $\nabla_L$ | Laplace operator, |
| $c$ | Specific heat capacity of gear steel materials |
| $\rho_m$ | Density of gear steel materials |
| $\lambda$ | Thermal conductivity |
| $T$ | Gear temperature |
| $Q$ | Joule thermal power density |
| $k$ | Heat transfer coefficient between gear surface and air |
| $T_{in}$ | Ambient temperature |
| $n$ | External normal to gear surface |
| $\varepsilon_a$ | Gear surface emissivity |
| $\sigma_a$ | Stefan-Bozeman constant |

**Abbreviations**

| | |
|---|---|
| TC | Transverse coil |
| LC | Longitudinal coil |
| GSPA | Gear shot peening area |
| GSTD | Gear surface temperature difference |
| TT | Tooth top |
| TR | Tooth root |

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
