# Peer review of "Dynamic Induction Heating Temperature Field Analysis of Spiral Bevel Gears"

_applsci, doi:10.3390/app121910018_

Round 1

Reviewer 1 Report

The authors have provided a well structured and interesting paper. The introduction is very good and outlines the necessary. However, the novelty of the paper is not explicitly explained. Below are some improvement suggestions the Reviewer kindly asks the authors to consider and amend the manuscript accordingly.

Major:

1)      The abstract and conclusion explains the core research and results superbly, however the readers are still left wondering what this research claim does to be novel.

a.       The abstract and conclusion should be able to reflect this and not just the introduction.

2)      Since the novelty is not very clear, below are some follow up questions.

a.       Can the authors provide explanation on the consequence of the study, in regard to, the gears operating in its desired environment.

b.       It appears only the pinion is considered for this study, a brief explanation of why the wheel was not considered would be beneficial for the reader.

3)      Although COMSOL can be considered very comprehensive, an explanation of fundamental theory used to model this investigation would be needed.

a.       The authors do not need to go into too detail, but the name and the written form of the main equations considered for the analysis should be presented.

b.       This would allow the reader to understand the nature of what factors are being considered for the analysis – even those not familiar with COMSOL or the physics.

c.       This would also enforce the explanations provided in the current result section where there are some differences between the experimental and simulated results.  

                                                               i.      Please add a nomenclature section at the end.

Minor

The following points are not necessary to change and up to the authors preference, however, are included as form of courtesy:

·         Some standalone figures such as Figures 13 and 14 are left justified, it would probably be more presentable if centre justified.

·         Figure 3, 17 and 18 appears to have been left justified too much – it might be better to split the figure up or make it marginally smaller.

Reviewer 2 Report

The article seems interesting due to the area of application: induction heating of spiral bevel gears of great importance in the automotive industry. The study was focused on two possible induction coils: a transverse and a longitudinal. The authors claim to have demonstrated the superiority of the longitudinal coil. It is very valuable that the authors have verified their numerical simulations with a physical experiment.

The Introduction chapter is comprehensive. The second chapter is in fact a classic "Materials and methods" describing a bevel gear under study, induction coil modeling and induction heating modeling. It should be noted that a bevel gear material is not described however all details of its geometry are considered. A COMSOL software was used to heating modeling.

The third chapter contains results and discussion of numerical simulations. Several combinations of outcomes and controlled factors were considered in the numerical simulation. The first combination wase gear shot-peening area and surface temperature difference in relation to current intensity and current frequency. The second combination was heating rate of the gear shot-peening area and gear tooth surface temperature distribution in relation to coil structure.

The fourth chapter is devoted to the subject of a physical experiment. This chapter is basically a mini-article with its own description of the methodology and analysis of the results. It was not stated whether the results of the physical experiment were obtained from a single test or whether the experiment had more replications. A separate subsection is the comparison of the results of the numerical simulation and the physical experiment. Unfortunately, this comparison is mainly qualitative (graph), while quantitative measures (max, average) are given without estimating the uncertainty.

Conclusions are clearly formulated and justified by presented results. The references contain 22 items: 8 from the last three years and 14 from previous decade. All items are adequatly selected.

The issue discussed in the article is of great engineering importance, but there are no significant scientific aspects. It is a correct and purposeful application of known computational methods verified by a physical experiment.

Remarks:

1. The materials of the bevel gear should be described in detail.

2. Physical test should be described more precisely. Whether the test was performed individually or were replications. What were the uncertainties in the measurements of the physical experiment.

Reviewer 3 Report

Although the topic is very interesting, the paper needs extensive improvement.

1. Were current values the same for both coils?

2.  Points 3,1 and 3,2 - it is not clear what are the values of individual quantities for the simulations, please add all parameters of the simulations: I, f, t. 

3. There is no mathematical model in the paper.

4. It is not clear for which gear point the temperature is reported. (Fig. 6 and 7).

5. lines 185-187 - As for the GSTD, when heated with a TC with a current of 1500 A, the maximum GSTD is 145 ° C. The GSTD is approximately 75 ° C when heated with an LC. These values do not correspond to the data shown in Figure 6.

6. lines 207-211 At the same current frequency, the temperature in the GSPA was approximately 50 °C higher with a TC than with an LC. When the TC was used for heating with a current frequency of 20 kHz, the temperature difference on the gear surface reached a maximum of 102 °C, whereas when the LC was used for heating, the temperature difference on the gear surface was approximately 50 °C. - Please provide temperature values.

7. Fig. 8 - Why different values on the X axis are assumed? Please insert the same ranges.

8. Line 240 wrong No of Figure.

9. Please explain the differences between the electromagnetic heat presented in figs. 13 and 14 and the temperature distribution presented in figs. 10 and 11.

10. Please explain the sentences:

lines 160-164 The induction heating process only solves the temperature field of the spiral bevel gear and the boundary conditions considering the thermal convection and thermal radiation. The convective heat transfer coefficient was set to 10 W/(m·K), radiation rate was 0.7, and external ambient temperature was set to 293 K.

lines 398 - 402: ... This is because only the temperature change of the gear itself is considered in the modeling process. The heat exchange between the gear and the surrounding air during the heating process and the radiation of the tooth surface temperature to the induction coil, which causes a change in the physical properties of the coil, are not considered. This affects temperature stability during the experiment.

11. Please explain how the electrical parameters in the coil, current and frequency were measured.

Round 2

Reviewer 1 Report

The Reviewer gives full appreciation to the authors effort in answering the questions. The manuscript is ready for publication.

Reviewer 3 Report

Accept in present form